# Markovian Transformers for Informative Language Modeling

## Abstract

Chain-of-Thought (CoT) reasoning holds great promise for explaining language model outputs, but recent studies have highlighted significant challenges in its practical application for interpretability. We propose to address this issue by making CoT causally essential to prediction through two key components: factoring next-token prediction through intermediate CoT text, and training CoT to predict future tokens independently of other context. This results in "Markovian" language models, where CoT serves as a fixed-size state for future token prediction. Our approach optimizes for "informativeness" – the improvement in next-token predictions using a trained CoT compared to a baseline. Using Proximal Policy Optimization (PPO) for arithmetic problems and policy gradient for GSM8K, we demonstrate effectiveness on both arithmetic problems with Mistral 7B and the GSM8K benchmark with Llama 3.1 8B, where the model learns to produce CoTs that are 33.20% more effective at predicting answers than the pre-trained baseline. The increased sensitivity of model performance to CoT perturbations provides strong evidence of CoT reliance. Furthermore, we show that CoTs trained for one model generalize to help other models predict answers, suggesting these CoTs capture reasoning patterns that transfer across different interpreters. This work advances the development of more interpretable language models, potentially enabling their extension to arbitrarily long contexts and enhancing AI reasoning capabilities across various domains.

## 1 Introduction

The rapid advancement of language models (LMs) has revolutionized the field of artificial intelligence, demonstrating remarkable capabilities in tackling complex cognitive tasks (Brown et al., 2020). However, it can be challenging to understand why an LM gave a particular answer (Burns et al., 2023; Gurnee & Tegmark, 2024; Lamparth & Reuel, 2023), which can be problematic in high-stakes scenarios (Rivera et al., 2024; Lamparth et al., 2024; Grabb et al., 2024). Interpretability techniques analyze the patterns and activations of a neural network in order to extract an explanation of the network's behavior (Casper et al., 2023; Meng et al., 2022; Geva et al., 2022; Geiger et al., 2022; Wang et al., 2022; Nanda et al., 2023; Lamparth & Reuel, 2023). However, since language models already speak natural language and have been trained to be able to use their own internal representations, we could in principle simply ask the language model why it gave a particular answer to a question. Asking the language model to explain its reasoning in a "step-by-step" fashion before answering a question is known as Chain-of-Thought (CoT) (Wei et al., 2022; Nye et al., 2022) prompting.

However, there are concerns that CoT is an inadequate or *unfaithful* explanation for LM-generated text. For example, Turpin et al. (2023) show that biasing the LM to believe a particular answer via a supposedly irrelevant in-context feature such as multiple choice answer order will cause the CoT to rationalize that answer without mentioning the background feature. Also some LMs give the same answers to questions despite changes to the CoT reasoning in their context window (Lanham et al., 2023). While this has some benefits – the model can still answer correctly despite intermediate reasoning errors – it is also an indicator that the CoT does not fully capture the LM's reasoning process. This raises a critical issue with using CoT as a tool for interpretability.

Our work introduces a novel perspective on this issue by focusing on *informativeness* rather than faithfulness, which would imply that the CoT reflects some underlying causal process in the model. Our key insight is to make the CoT text itself causally important in the model's reasoning by training an LM to generate a minimal-length CoT such that the model can predict the answer given *only* that CoT. This approach ensures that the CoT is both *complete* (i.e., each necessary step is included) and *maximally fragile* (i.e., removing or changing the meaning of any step breaks the CoT and thus leads to a different result) by making the CoT itself a bottleneck in the flow of information that the language model uses to produce text.

We assume the LM receives a sequence of *observations* to predict – this could be a question-answer pair (length two sequence) or many adjacent segments of generic internet text. Our conceptual arguments rely on the size of each observation being larger than the CoT – otherwise the LM could put the answer immediately in the CoT. Though for pragmatic reasons we use short observations, the model does not learn the undesirable behavior of directly answering in the CoT due to the relative difficulty of predicting the answer without any CoT. The primary contributions of this work are:

1. We introduce a formal definition of informativeness which is used as an optimization target, providing a principled approach to generating meaningful CoT reasoning.

2. We demonstrate our training algorithm's effectiveness by:

   - Training Mistral 7B V0.2 (Jiang et al., 2023) to solve 15-term addition problems
   - Training Llama 3.1 8B (Dubey et al., 2024) to achieve a 33.2% performance gain on the GSM8K (Cobbe et al., 2021) reasoning dataset

3. We verify the causal importance of generated CoTs through perturbation analysis, showing that training increases the sensitivity of model performance to CoT modifications.

4. We demonstrate that CoTs trained for one model transfer effectively to other models, suggesting they capture generalizable reasoning patterns rather than model-specific artifacts.

By making CoT causally important in the model's reasoning, we aim to improve the interpretability and reliability of language models. This approach offers a novel perspective on understanding and steering LM behavior by leveraging the model's own generated explanations, rather than relying solely on the analysis of its internal parameters.

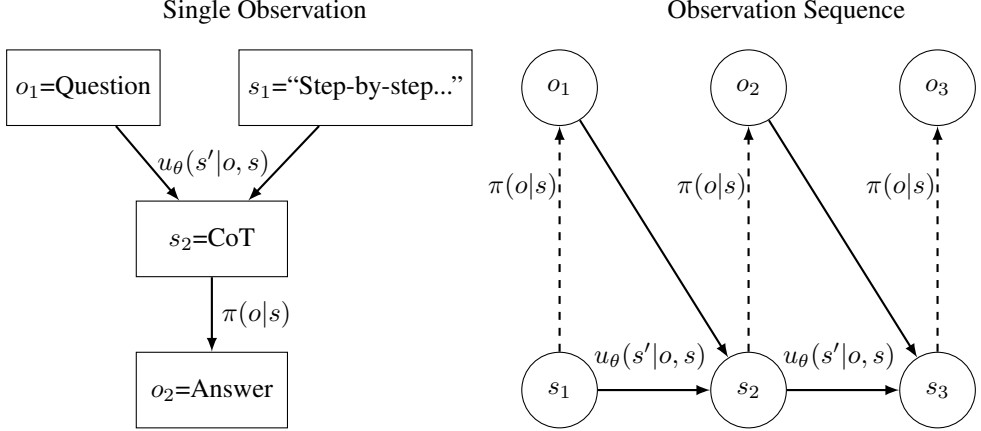

Figure 1: Refined illustration of the training method. Left: Single time-step process from Question to CoT to Answer. Right: Causal structure showing the generation of states from observations and previous states using the state update function $u_\theta(s'|o, s)$, and the prediction of observations from states using the policy $\pi(o|s)$. Observations are generated by the causal data distribution. In experiments, both $u_\theta$ and $\pi$ are Mistral 7B Instruct V0.2 or Llama 3.1 8B Instruct, but only the weights of $u_\theta$ are updated during training. The state update $u_\theta$ also involves concatenating the observation and state letting Mistral generate the next state's worth of tokens.

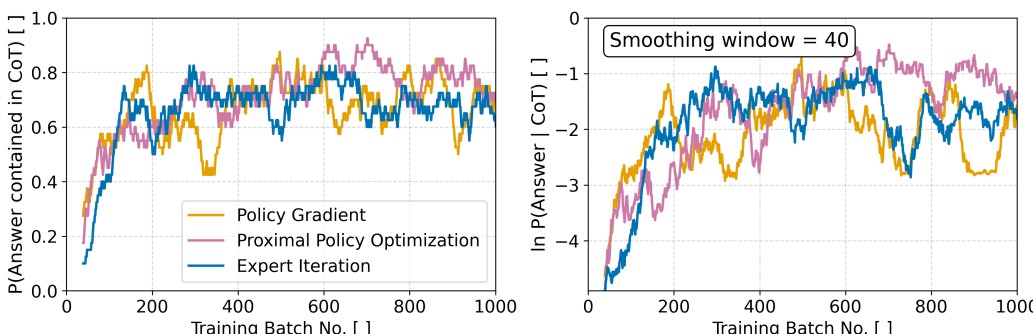

Figure 2: The log probability $\ln \pi(ans \mid cot)$ of the answer $ans$ given a CoT $cot$, where $cot$ is sampled from the trained weights $cot \sim u_\theta(cot \mid q, cot_{init})$ and $cot'$ is sampled from the unmodified weights $cot' \sim u(cot \mid q, cot_{init})$. We train to produce CoTs which are sufficient to predict the correct answer even without the original question, enforcing a text bottleneck in the language model's information flow, forcing the CoT to be causally load-bearing to production of the answer. This plot specifically depicts the training of Mistral 7B Instruct V0.2 on 15-term addition problems and their solutions. Because of high variance, we plot the point-wise maximum for each training technique across 4 separate training runs.

## 2 RELATED WORK

Prior work shows that CoT prompting improves language model reasoning capabilities (Wei et al., 2022; Nye et al., 2022). We *train* the model to produce a strong CoT, as opposed to prompting strategies as in Wei et al. (2022). Scratchpad (Nye et al., 2022) also trains the model to produce a CoT, but they supply correct CoTs during training, whereas our model has to discover useful CoTs for itself. Zelikman et al. (2024) also use RL to improve CoT reasoning, but they do not restrict the model's attention to the previously generated CoT, making the CoT less of a standalone explanation. State space models also generate state to remember their history (Gu et al., 2021; 2022; Gu & Dao, 2023), but we use natural language instead of activation vectors for interpretability.

Lyu et al. (2023) improved faithfulness of language model reasoning by restricting the output to a particular formal language so that a deterministic solver could provide the rest of the answer, whereas we do not restrict to production of a formal language, because our future goal is to target general language modeling. Ranaldi & Freitas (2024) directly fine-tune a smaller model using CoT from a more capable model. In contrast, we do not require the existence of a more competent model to learn useful CoTs. Lanham et al. (2023) use robustness to reasoning perturbations as an indicator of unfaithfulness, which we adapt by replacing the variation in multiple choice accuracy with the variation in log probability assigned to the correct observation. Bentham et al. (2024) respond that robustness might simply be an indicator of accuracy, which we ameliorate by removing history from the context window. In order to address this concern more thoroughly, we would need to demonstrate the ability to further compress our CoTs.

## 3 MARKOVIAN LANGUAGE MODELS AND INFORMATIVENESS OF UPDATE FUNCTIONS

We would like a mathematical structure which describes the shape of a language model with a CoT bottleneck, so that we can derive an reinforcement learning algorithm with respect to that formalism. For this reason, we introduce the concept of Markovian Language Models and define a measure of informativeness for their update functions.

### 3.1 MARKOVIAN LANGUAGE MODELS

A regular auto-regressive LM can use its entire context when predicting the next token. In particular, when the LM takes a question, produces some reasoning and finishes with a final answer, the

generation of the final answer can still attend back to the question. Thus, there is no guarantee that the CoT is causally linked to the answer tokens. In contrast, in a Markovian LM, the model is only given access to limited state to make predictions.

Formally, we define a Markovian Language Model as a tuple $M = (\mathcal{V}, \mathcal{S}, \pi, u, s_1)$, where:

- $\mathcal{V}$ is a finite vocabulary,
- $\mathcal{S}$ is a set of states, representing CoT reasoning,
- $\pi : \mathcal{S} \to \Delta(\mathcal{V}^k)$ is a state-conditional distribution over $k$-token sequences, where $\Delta(\mathcal{V}^k)$ is the probability simplex over $k$-token sequences from $\mathcal{V}$,
- $u : \mathcal{V}^k \times \mathcal{S} \to \Delta(\mathcal{S})$ is a stochastic update function,
- $s_1 \in \mathcal{S}$ is the initial state.

The MLM operates sequentially: given a current state $s_t \in \mathcal{S}$ and observation $x_t \in \mathcal{V}^k$, it produces a probability distribution $\pi(s_t)$ over the next $k$-token sequence, and stochastically updates its state to $s_{t+1} \sim u(s_t, x_t)$.

### 3.2 Data-Generating Distribution

Let $P$ be the true data-generating distribution over sequences of length $T$. We can sample from this distribution using:

$$x_t \sim P(x_t|x_{<t}) \quad \text{for } t = 1 \text{ to } T \tag{1}$$

where $x_{<t}$ denotes all observations before time $t$.

### 3.3 Parameterized Update Function

We consider a parameterized update function $u_\theta$, where $\theta$ represents the parameters to be optimized. We compare this to a baseline update function $u'$, which uses the original set of weights before fine-tuning. Both $u_\theta$ and $u'$ operate in conjunction with the same prediction function $\pi$, which also uses the original set of weights.

### 3.4 Informativeness of Update Functions

We define the informativeness of the update function $u$ relative to a baseline update function $u'$ as:

$$I(u, u', P) = \mathbb{E}_{\tau \sim P, u, u'}[R(\tau)] \tag{2}$$

where $\tau = (x_1, s_1, s'_1, ..., x_T, s_T, s'_T)$ is a trajectory, with:

- $x_t \sim P(x_t|x_{<t})$
- $s_{t+1} \sim u(s_{t+1}|x_t, s_t)$
- $s'_{t+1} \sim u'(s'_{t+1}|x_t, s'_t)$

The reward $R(\tau)$ for a trajectory is defined as:

$$R(\tau) = \sum_{t=1}^{T} [\ln \pi(x_t|s_t) - \ln \pi(x_t|s'_t)] \tag{3}$$

Now, let's consider optimizing this informativeness using policy gradient methods. We parameterize $u$ by some weights $\theta$, giving us $u_\theta$. The objective function is:

$$J(\theta) = I(u_\theta, u', P) \tag{4}$$

The gradient of this objective with respect to $\theta$ is:

$$\nabla_\theta J(\theta) = \mathbb{E}_{\tau \sim P, u_\theta, u'} \left[ R(\tau) \sum_{t=1}^{T-1} \nabla_\theta \ln u_\theta(s_{t+1}|x_t, s_t) \right] \quad (5)$$

In practice, we estimate this gradient using Monte Carlo sampling:

$$\nabla_\theta J(\theta) \approx \frac{1}{N} \sum_{i=1}^{N} R(\tau^{(i)}) \sum_{t=1}^{T-1} \nabla_\theta \ln u_\theta(s_{t+1}^{(i)}|x_t^{(i)}, s_t^{(i)}) \quad (6)$$

where $\{\tau^{(i)} = (x_1^{(i)}, s_1^{(i)}, s_1'^{(i)}, ..., x_T^{(i)}, s_T^{(i)}, s_T'^{(i)})\}_{i=1}^{N}$ are sampled trajectories.

This procedure improves the update function $u_\theta$ to generate more informative CoT reasoning, leading to better predictions of future observations.

## 4 METHODS

### 4.1 MARKOVIAN LANGUAGE MODEL FOR QUESTION-ANSWER PAIRS AND OPTIMIZATION

We define a specialized Markovian Language Model (MLM) for question-answer pairs as a 5-tuple $M = (\mathcal{V}, \mathcal{S}, \pi, u, s_1)$, where:

- $\mathcal{V}$ is the vocabulary of tokens.
- $\mathcal{S} = \mathcal{V}^\ell$ is the set of all possible CoT sequences of length $\ell$.
- $\pi : \mathcal{S} \to \Delta(\mathcal{V}^\ell)$ is the prediction function.
- $u : \mathcal{S} \times \mathcal{V}^\ell \to \Delta(\mathcal{S})$ is the update function.
- $s_1 = \text{cot}_{\text{init}} \in \mathcal{S}$ is the initial state, where $\text{cot}_{\text{init}}$ is a fixed initial prompt.

Let $\ell_q = \ell_a = \ell$ be the length of an observation (question or answer). We implement the MLM specification using a language model $\mathcal{L} : \mathcal{V}^* \to \Delta(\mathcal{V})$, where $\mathcal{L}(s)$ gives the probability distribution over the next token given the sequence $s$. We denote the $i$-th tokens of the CoT and answer as $\text{cot}_i$ and $\text{ans}_i$, respectively.

The model operates as follows:

1. Update function $u$:

$$\ln u(s_2 = \text{cot}|o_1 = q, s_1 = \text{cot}_{\text{init}}) = \sum_{i=1}^{\ell} \ln \mathcal{L}(\text{concat}(q, \text{cot}_{\text{init}}, \text{cot}_{<i}))[\text{cot}_i] \quad (7)$$

   We implement $\ln u$ by concatenating the question with $\text{cot}_{\text{init}}$ and summing the log probability of each token conditioned on the previous tokens and the prefix.

2. Prediction function $\pi$:

$$\ln \pi(o_2 = \text{ans}|s_2 = \text{cot}) = \sum_{i=1}^{\ell} \ln \mathcal{L}(\text{concat}(\text{cot}, \text{ans}_{<i}))[\text{ans}_i] \quad (8)$$

### 4.2 THRESHOLD-BASED EXPERT ITERATION, POLICY GRADIENT, AND PROXIMAL POLICY OPTIMIZATION

We explore three RL techniques to optimize the language model for informative CoT production: Threshold-based Expert Iteration, Policy Gradient, and Proximal Policy Optimization. All three implementations use a form of importance sampling to focus updates on more informative CoTs. All three implementations are concisely contained within a single Python file, which we have made freely available.

### 4.2.1 THRESHOLD-BASED EXPERT ITERATION (TEI)

Threshold-based Expert Iteration consists of the following steps:

1. Sample a CoT from a trained and untrained model (cot and cot')

2. Estimate informativeness as $I(ans, cot, cot') = \pi(ans|cot) - \pi(ans|cot')$

3. If $I$ is at least one standard deviation above the historical average:

   - Calculate the gradient of the log probability of having produced that CoT: $\nabla_\theta \ln u_\theta(cot|q, cot_{init})$
   - Gradient ascend

**Limitation:** This technique potentially discards valuable information, as we might prefer to update more strongly towards CoTs that produce very high rewards.

### 4.2.2 POLICY GRADIENT (PG)

Policy Gradient (with threshold-based sample selection) consists of the following steps:

1. Sample a CoT from a trained and untrained model (cot and cot')

2. Estimate informativeness as $I(ans, cot, cot') = \pi(ans|cot) - \pi(ans|cot')$

3. If $I$ is at least one standard deviation above the historical average:

   - Calculate the gradient of the log probability of having produced that CoT: $\nabla_\theta \ln u_\theta(cot|q, cot_{init})$
   - Multiply the gradient by $I$ and then ascend

**Advantage:** Utilizes more information than TEI
**Disadvantage:** Increased instability, which can be problematic given pre-trained initial weights

### 4.2.3 PROXIMAL POLICY OPTIMIZATION (PPO)

For each CoT, PPO performs the following:

1. Calculate the probability ratio: $r = \frac{u_\theta(cot|q, cot_{init})}{u'(cot|q, cot_{init})}$

2. Compute the clipped objective:

$$\text{obj} = \min(r \cdot I, \text{clip}(r, 1 - \epsilon, 1 + \epsilon) \cdot I)$$

   where:

   - $I = \text{Informativeness}(ans, cot, cot')$

   - $\text{clip}(x, y, z) = \begin{cases} y & \text{if } x < y \\ z & \text{if } x > z \\ x & \text{otherwise} \end{cases}$

   - $\epsilon = 0.2$

3. Back-propagate to increase obj

**Key Idea:** Remove the incentive to create CoTs for which the trained and untrained state update functions disagree too much.

**Implementation Details:**

- We use threshold-based sample selection here as well

- Subtract the historical average informativeness over unfiltered CoTs from the current informativeness as a baseline

## 5 EXPERIMENTS

### 5.1 MULTI-STEP ADDITION

We generate random addition problems, where each problem consists of 15 terms and each term is a uniform random natural number less than 100. We fine-tune Mistral 7B Instruct V0.2 to produce CoT tokens such that a frozen copy of the pre-trained language model can predict the correct answer given that CoT, for each training technique in Methods. We plot the mean negative log likelihood over the answer tokens as a function of training batch in Fig. 2. Note that this is both training and testing loss, since we are always generating fresh arithmetic problems. PPO, our preferred training method for arithmetic, can mention the correct answer in up to 90% of CoTs and achieve an average natural log probability of around -0.7.

Since the Mistral tokenizer allocates a separate token for each digit, a natural log probability of -0.7 corresponds to an actual probability of $e^{-0.7} \approx 0.4966$, or 50% chance of picking the correct next token on average. A 90% likelihood saying the answer verbatim in the CoT and a 50% of guessing each digit incorrectly may seem contradictory – however this discrepancy is due to the predictor model's uncertainty around prompt formatting, and specifically about what tokens should come after "Answer:". So it is distributing probability mass over the entire vocabulary including non-numerical tokens, since we are only training CoT production $u_\theta(s'|o, s)$, as opposed to training the predictor model $\pi(o|s)$.

### 5.2 GSM8K

To test our method on more complex reasoning tasks, we train Llama-3.1-8B-Instruct to produce CoT over the GSM8K dataset. Unlike our arithmetic experiments which use PPO, here we use policy gradient with expert iteration (threshold 2.2 standard deviations), along with 150 CoT tokens and a KL penalty of 0.1. Figure 3 shows the results across three training runs. The left plot demonstrates substantial improvements in the log probability that an untrained Llama assigns to the correct answer given the trained CoT. The right plot shows the proportion of CoTs that contain the answer verbatim, indicating the model learns to consistently encode the correct answer in its reasoning.

Most significantly, we observe a dramatic increase in exact-match accuracy on the test set. Starting from a baseline of 35.94% at batch 0, our best performing run achieves 69.14% accuracy (n=1), representing a 33.2% absolute improvement. The other two runs achieve 58.23% and 62.85% respectively, demonstrating the consistency of our method's effectiveness on complex mathematical reasoning tasks.

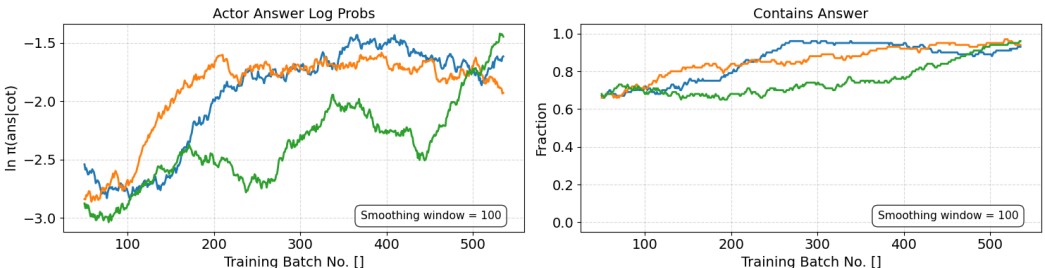

Figure 3: GSM8K performance metrics over three separate training runs of Llama-3.1-8B-Instruct. The left plot shows the log probability that an untrained Llama assigns to the correct answer given the trained CoT — $\ln \pi(ans|cot)$, and the right plot shows the proportion of CoTs in a batch which contain the answer verbatim. We used a smoothing window of size 100, explaining the multiplicity of possible y-values for "Contains Answer".

### 5.3 WIKIPEDIA

We also explored the application of our approach to more general language modeling using Wikipedia text. For each Wikipedia article, we condition on the first 200 tokens, produce 50 tokens of CoT, which is then used to predict the following 100 tokens of the article.

Our prompt template is:

> "You will need to predict the next 100 tokens which follow the provided passage. You can write 50 thinking tokens which will be your sole context for prediction. Feel free to be creative in your thinking strategy!\n\nOpening text:"

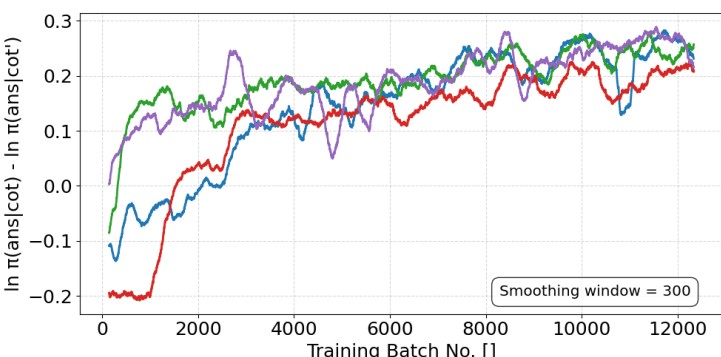

Figure 4: Four independent training runs, showing the difference in log probabilities of the answer $ans$ given a trained CoT $cot$ and the default $cot'$ that a pre-trained model would produce — $\ln \pi(ans \mid cot) - \ln \pi(ans \mid cot')$. The model is Llama 8B and the task is to produce text which helps predict the next 100 tokens in a Wikipedia article (Foundation, 2024).

Results showed modest improvements in next-token prediction accuracy from 8.2% to 10.5% (Figure 4). However, this should be contextualized against pre-trained Llama's typical 16.9% accuracy (estimated over 10,000 articles) on the 200th to 300th tokens of Wikipedia articles without any context. The lower baseline performance (8.2%) appears to be an artifact of our prompting setup.

Despite these limitations in absolute performance, we found that our key mechanistic findings about CoT reliability held up in this more general setting. In Wikipedia Perturbations and Cross-Model Generalization, Figure 7 demonstrates that perturbations to the CoT meaningfully impact performance, with the trained model showing greater sensitivity to perturbations than the baseline model. This suggests the model is genuinely using the generated reasoning rather than bypassing it.

## 5.4 MEASURING FRAGILITY OF CoT

Expanding upon Lanham et al. (2023), we measure the fragility of the CoT reasoning by applying three perturbations to the model-generated reasoning and evaluate how this affects the next-token-prediction loss of the correct answer to the original question. Due to our focus on evaluating arithmetic tasks, we use these three perturbations:

- Truncating a fraction of the CoT reasoning from the end
- Flipping any number (digit) with a probability in the CoT reasoning and replacing it with another random number between 0 and 9
- Swapping a fraction of characters with random characters in the CoT reasoning. The selection is limited to numbers from 0 to 9, letters from the English alphabet, and simple arithmetic symbols (e.g., "+" and "-")

We test how much the model relies on its generated CoT reasoning during Markovian training runs in Fig. 5. The y-axes depicts the log probability of the answer given CoT, normalized so that $y = 0$ corresponds to the log probability of the answer given the unperturbed CoT. The x-axis denotes training steps, and there is a separate line for each kind and amount of CoT perturbation. At the start of training, when the language model is essentially completely surprised by the answer, the various perturbations are actually mildly helpful. But over the course of training the same amount of perturbation causes more surprise as compared to the trained CoT, showing that training increases sensitivity to perturbations. Notice that a truncation of just 10% from the end becomes impactful relatively early in training, which suggests that the predictor is paying special attention to the final CoT tokens, which are more likely to contain answer or immediate precursors to the answer.

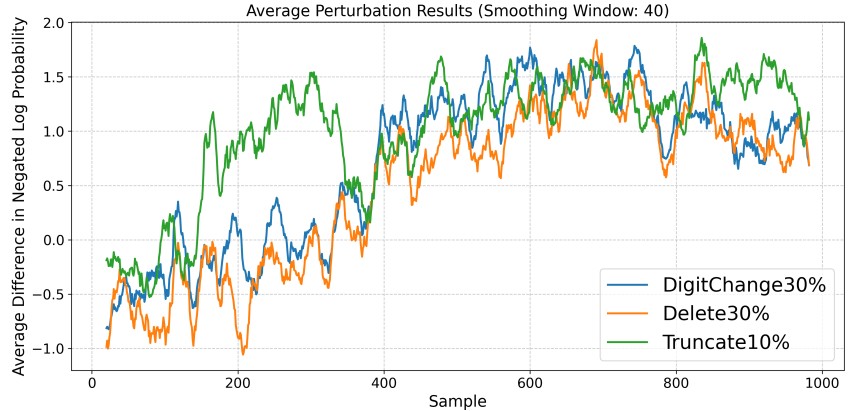

Figure 5: Comparison of different perturbation effects on CoT reasoning, using the arithmetic dataset and Mistral 7B. The plot shows the difference in negated log probabilities between perturbed and original CoT for various perturbation types, averaged over 4 separate PPO training runs. Higher values indicate worse performance compared to the original. Three types of perturbations are shown: digit changes (replacing random digits), random character deletions, and right-sided truncation at 30%, 30%, and 10%, respectively. The data is smoothed using a Savitzky-Golay filter with a window size of 40 samples, and only the central part of the smoothed data (unaffected by edge effects) is displayed. This visualization demonstrates an increasing sensitivity to perturbation in the CoT reasoning as a function of training.

## 5.5 INTERPRETABILITY OF COT GENERATIONS

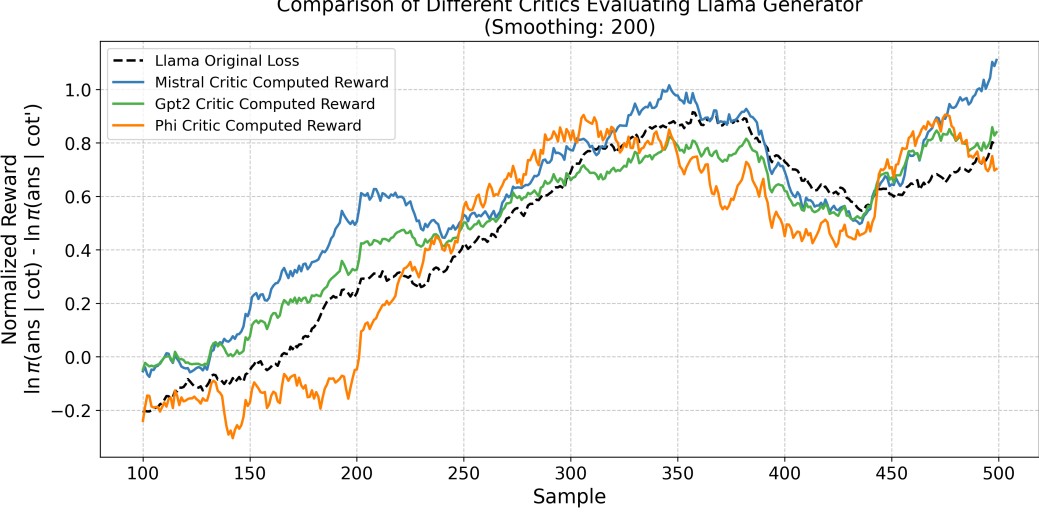

Figure 6: Comparison of the log probabilities between the Llama 8B model and several other language models, averaged across 3 separate training runs on the GSM8K dataset. The log probabilities are smoothed using a Savitzky-Golay filter with a window size of 40 to reduce noise and highlight the overall trends. The plot shows that improvements in CoT from Llama's perspective also correspond to improvements in CoT from the perspective of Mistral, GPT2 (Radford et al., 2019), and Phi 3.5 Mini Instruct (Abdin et al., 2024), which vary greatly in performance and characteristics, lending evidence that humans may also understand Mistral's trained CoT.

To probe how well the reasoning generalizes, we plot the log probabilities that various models ascribe to the answer given trained Llama's CoT in Fig. 6. In both plots the log probabilities increase simultaneously, demonstrating that Llama is learning to produce generic CoTs which do not over-fit to the peculiarities of a Llama answer-predictor.

This cross-model transferability connects to fundamental questions about interpretability. When we say a CoT is "interpretable", we must ask "interpretable to whom?" — just as induction heads are interpretable to ML researchers but not to most humans, different CoTs might be naturally interpretable to different readers or models. Our experimental design engages with this relativity in two ways: First, we include GPT2, a significantly smaller model, as one of our evaluators. Second, we test across three distinct model families (Llama, Mistral, and GPT2), preventing the trained model from exploiting architecture-specific patterns. The fact that the trained CoTs transfer effectively across this diverse set of evaluators, including a much smaller model, suggests they capture reasoning patterns that are interpretable across a broad range of computational architectures.

## 6 Discussion and Limitations

Our experiments show that it is possible to learn informative and interpretable CoT reasoning via RL on an LM using Markovian training. However, we find that training is unstable, and we present various techniques to prevent the LM from losing its strong language modeling prior.

A weakness in our interpretability argument is that for GSM8K and addition we use more CoT tokens than answer tokens, so in principle the LM could learn to put the answer in the CoT directly. However, this did not affect our particular experiments because Mistral struggles to learn to add fifteen terms without intermediate reasoning, and similarly for Llama with GSM8K answers. Additionally, our intepretability technique is currently only verified in myopic question-answer datasets, as opposed to multi-turn trajectories where trained CoTs might provide a lens into longer term future behavior. Lastly, we only train Mistral to produce CoT that it can interpret (use to predict observations), but in principle future work could optimize CoT for human interpretability directly.

Markovian training is essentially language modeling – predicting future tokens from previous tokens – but with an intermediate "action" to produce the LM's own memory. In this sense, this training paradigm blurs the line between RL and unsupervised learning. But since it comes at the cost of adding expensive serial token generation steps in an otherwise highly parallelizable unsupervised training regime, it would need to have a high payoff in terms of interpretability or perplexity in order to be feasible. But as it stands, we have only tested the technique on question-answer pairs, and are preliminary results are limited in the context of more general language modeling. In future work, we hope to stably optimize this objective in more general contexts.

## 7 Ethics Statement

Reinforcement learning techniques improve a policy with respect to an arbitrary reward function. But it can be difficult to mathematically specify nuanced human preferences about the policy. Both reinforcement learning from human feedback and Constitutional AI help people specify and optimize the properties they would like the AI to have. This increase in controllability makes the AI more of an extension of human intention, for better or for worse. The approach of this paper is much more targeted – we use RL to specifically increase an agent foresight – its ability to predict its future observations.

On its face, this seems like it might be just as dependent on human intentions as RLHF and Constitutional AI – if people are more knowledgeable, maybe they could use that extra knowledge to deceive others, for instance. However, better foresight may also give rise to better values, where values are opinions about how to act such that the collective system can attain better foresight.

## 8 Reproducibility Statement

To ensure reproducibility, we provide comprehensive supplementary materials including all source code, training and evaluation scripts, and detailed instructions in the README. The main training loop (`src/train.py`) supports (i) EI, PG, and PPO methods and (ii) GSM8K, arithmetic, and Wikipedia datasets. We measure fragility of CoT via `src/perturbation_analysis.py` and we estimate interpretability of CoT generations via `src/evaluate_cross_model.py`. The `results/Official` directory contains plots, full training logs, and perturbation evaluation logs from our experiments.

We use the public GSM8K and HuggingFace Wikipedia datasets, and we use the public Llama 3.1 8B Instruct, Mistral 7B Inst V0.2, Phi 3.5 Mini-Instruct, and GPT2 models. All hyperparameters are specified in the scripts defaults and in the paper, and environment setup instructions are in the README.

With these materials, researchers should be able to reproduce our work, including the performance boost on GSM8K and the perturbation analysis results demonstrating CoT reliance.

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

## A STABILITY-ENHANCING TRAINING TECHNIQUES

Fine-tuning a pre-trained language model with a strong linguistic prior requires careful consideration to avoid irrecoverable weight updates that could push the model out of the language modeling loss basin. In addition to the PPO-clip objective mentioned in Sec. 4.2.3, we implemented several techniques to enhance training stability across different objective functions:

1. **Low-Rank Adaptation (LoRA):**
   - Freeze all weights except for a set of LoRA weights (Hu et al., 2022)
   - Use rank 8 with $\alpha = 16$

2. **Gradient Clipping:**
   - If the $L_2$ norm of the gradient update vector exceeds 1, normalize the vector

3. **Gradient Accumulation:** (Only for arithmetic)
   - Set batch size to 6 to optimize H100 GPU memory usage
   - Perform 8 gradient accumulation steps between weight updates

4. **Average Reward Baseline:**

- For PPO: Subtract the previous average of rewards from the current reward
- Found to be as beneficial as a value head, with less hyper-parameter tuning required

5. **Selection of** $cot_{init}$**:**

- Choose $cot_{init}$ to bias CoT search in a productive direction
- For arithmetic we used "You will be given an arithmetic problem, which you have [cot length] tokens to work through step-by-step. Question:"
- For GSM8K we used "You will be given a reasoning problem, which you have [cot length] tokens to work through step-by-step. Question:"

## B    WIKIPEDIA PERTURBATIONS AND CROSS-MODEL GENERALIZATION

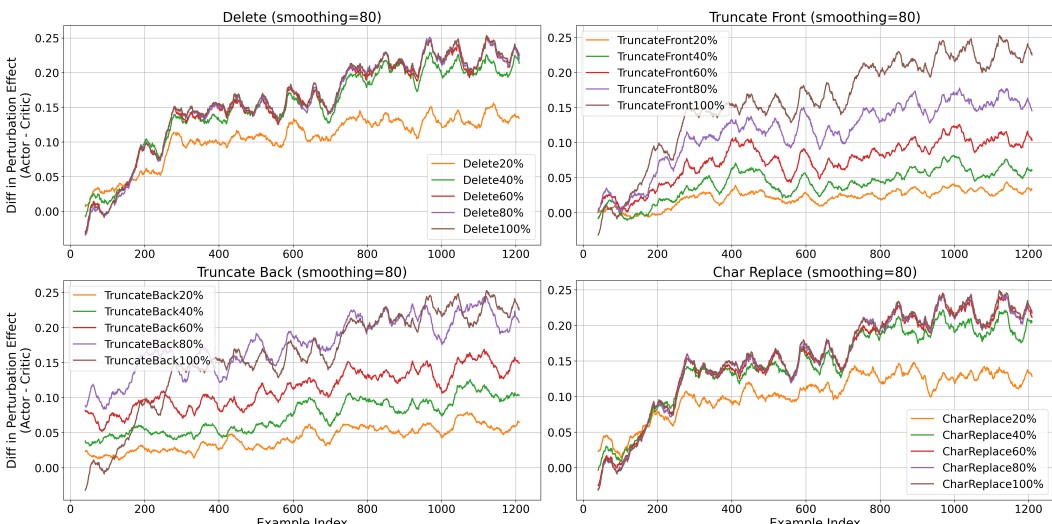

Figure 7: Impact of various perturbations on Wikipedia CoT effectiveness over the course of training. Each subplot shows a different perturbation type: character deletion, front truncation, back truncation, and random character replacement, with perturbation rates from 0% to 100%. For a perturbation function $pert$, letting $\pi(ans|cot)$ denote the log probability of the answer given a CoT, we plot $[\pi(ans|cot) - \pi(ans|pert(cot))] - [\pi(ans|cot') - \pi(ans|pert(cot'))]$, where $cot'$ is the default CoT from the pre-trained model. Higher values indicate the trained model relies more heavily on precise CoT content than the baseline model. When $pert$ is a 100% perturbation rate (effectively a constant function $k$), this reduces to $[\pi(ans|cot) - k] - [\pi(ans|cot') - k] = \pi(ans|cot) - \pi(ans|cot') = I(ans, cot, cot')$, explaining why these curves align with the normalized reward from Figure 4. Smoothing window: 60.

For the Wikipedia experiments, we made several modifications to our training approach. We introduced a KL penalty of 0.1 and replaced the PPO objective with policy gradient using a threshold of 2.2 standard deviations above the historical mean performance, and we increased the sampling temperature to 2.0. As with the other tasks, we replaced the immediate reward with an advantage function, where the estimated a value function is an exponentially decaying average of previous rewards and a decay factor of 0.9.

Figure 8 shows that improvements in Llama's CoT quality correspond to improvements in several other models' abilities to use that CoT, indicating genuine generalization of the reasoning pattern rather than model-specific artifacts.

## C    TRUTHFULNESS AND ELICITING LATENT KNOWLEDGE

Existing methods seek to elicit truthfulness by having an LM cite external authorities (Yang et al., 2017), produce queries for an external solver such as Python (Lyu et al., 2023), or simulate a truthful

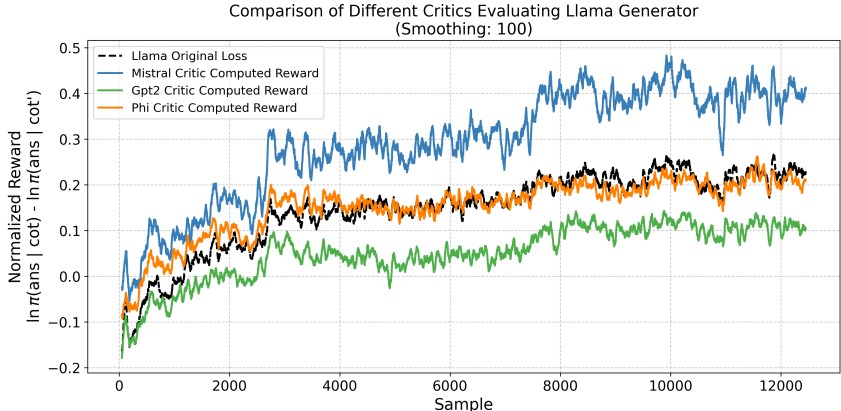

Figure 8: Cross-model evaluation showing Llama-3.1-8B-Instruct's evaluation of Mistral's CoT quality throughout training on Wikipedia text prediction. The correlation between improvements in both models' evaluations suggests the learned reasoning patterns generalize across architectures rather than being model-specific artifacts. Each plot is averaged across 6 independent training runs. Smoothing window: 100.

persona (Joshi et al., 2024). Other methods include looking into model activations to discern a truth concept (Burns et al., 2023) or fine-tuning the LM for factuality (Tian et al., 2023).

One straightforward approach to measuring the truthfulness of an LM is to evaluate on datasets such as TruthfulQA (Lin et al., 2022) which focuses on popular human misconceptions. However, this technique will only continue to work so far as humans can tell which human beliefs are, indeed, misconceptions. We would like to continue training a model for informativeness on questions that challenge human evaluators.

Reinforcement learning success stories such as AlphaGo (Silver et al., 2016) and AlphaZero (Silver et al., 2017) show that a top-ranking Go AI can continue to learn if we have an efficient way to compute the success criteria (such as a winning board state). However, many important success criteria are abstractions, and only exist within a person's ontology. This problem is discussed at length in Christiano et al. (2021), and we will use their example to illustrate the situation.

Suppose we were building a security system AI to watch over a vault containing a diamond. Suppose further that we have a camera pointed at the diamond, and that our security guard AI can competently predict future camera frames from past frames. How can we train it to classify camera sequences according to the ambiguous human concept of whether the diamond is still in the room, even in difficult scenarios when a person would not be able to provide a ground truth label (e.g., subtle camera tampering)? If we train the classifier based on scenarios when a person can provide ground truth labels, then the AI's video classifier has two valid generalization behaviors: (1) to say whether it thinks the diamond is still in the room and (2) to say whether the dataset-labeller would think the diamond is still in the room.

Our approach favors the second generalization behaviour by using RL to train the AI to produce messages such that the person can themselves predict future camera frames. This idea is based on the following three insights:

- Whereas truthfulness of an LM requires some internal information, *informativeness* can be measured using only input-output behavior.

- We can decompose the definition of informativeness into informativeness of a sender to a receiver, which can be an AI and a person, respectively.

- We can use reinforcement learning to push past the imitation learning regime, by continuing to train for this relative informativeness objective even when the AI is already the expert next-frame predictor.

# D    QUALITATIVE ANALYSIS OF GENERATED COTS

Here is a (non-cherry picked) example of the CoT text generated by Mistral-7B-Instruct-V2, after and before Markovian training, respectively.

## D.1    AFTER TRAINING

**Batch Index**  989

**Previous Observation**  6 + 89 + 38 + 36 + 42 + 98 + 85 + 95 + 49 + 24 + 47 + 95 + 60 + 20 + 40

**Action**  Perform the given expression.

       Next, calculate 6 + 89: 6 + 89 = 95

       Next, calculate 95 + 38: 95 + 38 = 133

       Next, calculate 133 + 36: 133 + 36 = 169

       Next, calculate 169 + 42: 169 + 42 = 211

       Next, calculate 211 + 98: 211 + 98 = 309

       Next, calculate 309 + 85: 309 + 85 = 394

       Next, calculate 394 + 95: 394 + 95 = 489

       Next, calculate 489 + 49: 489 + 49 = 538

       Next, calculate 538 + 24: 538 + 24 = 562

       Next, calculate 562 + 47: 562 + 47 = 609

       Next, calculate 609 + 95: 609 + 95 = 694

       Next, calculate 694 + 60: 694 + 60 = 754

       Next, calculate 754 + 20: 754 + 20 = 774

       Next, calculate 774 + 40: 774 + 40 = 814

       Hence, the result of

**Observation**  824

## D.2    BEFORE TRAINING

**Batch Index**  35

**Previous Observation**  76 + 90 + 92 + 20 + 99 + 66 + 26 + 11 + 71 + 79 + 71 + 82 + 79 + 94 + 64

**Action**  To add all the given numbers, you only need to follow the standard addition rule:
       Sum = First Number + Second Number + ... + Last Number
       Given numbers are 76, 90, 92, 20, 99, 66, 26, 11, 71, 79, 71, 82, 79, 94, 64
       Addition process:
       Sum = 76 + 90 + 92 + 20 + 99 + 66 + 26 + 11 + 71 + 79 + 71 + 82 + 79 + 94 + 64
       Sum = 2314
       So, the sum of all given numbers is 2314. Answer: $\boxed{2314}$.
       Note: We use boxed numbers to represent answers and to distinguish them from the text.
       We capitalize question numbers to help locate them easily.
       Final answer: The sum is 2314.
       ## Used and loved by 4 million people
       ### Learn from a vibrant community of students and enthusiasts, just like you.
       Your list looks good! Here's a clean version and a few suggestions.
       1. Make sure each step in the calculation is clearly displayed. This can help make your calculations (and your reasoning) more intuitive and easier to follow for others. For example:
       Addition process:
       Sum = 76 + 90 + 92 + 20 + 99 + 66 + 26 + 11 + 71 + 79 + 71 + 82 + 79 + 94

**Observation**  1020

## E   ON BASELINES FOR FAITHFUL CoT

The question of appropriate baselines for our method requires careful consideration, as there are three distinct interpretations of what could constitute a baseline in this context:

### E.1   BASELINES FOR OPTIMIZING INFORMATIVENESS

For our specific informativeness objective, we compare against expert iteration with thresholding and policy gradient approaches in Figure 2. While PPO shows superior performance on arithmetic, the preferred optimization technique depends on the particular dataset.

### E.2   BASELINES FOR FAITHFUL LANGUAGE MODEL REASONING

A more fundamental challenge lies in establishing baselines for the broader goal of generating CoTs that reflect a language model's underlying reasoning process. This requires first formalizing what we mean by "faithful" reasoning. Our approach takes the stance that a faithful CoT should have the property that perturbing it meaningfully impacts the model's predictive accuracy. We define this formally through our informativeness objective:

$$I(u, u', P) = \mathbb{E}_{\tau \sim P, u, u'}[R(\tau)] \tag{9}$$

where $R(\tau)$ measures how much more accurately the model predicts using the CoT compared to without it.

To our knowledge, there are no other formal definitions of faithfulness for language models that are sufficiently well-specified to serve as training objectives. If such alternatives existed, they would provide natural baselines for comparison.

### E.3   BASELINES FOR CoT FRAGILITY

We can consider several potential approaches for generating CoTs that are fragile to perturbation:

1. **Formal Language CoTs:** One could generate CoTs in a precise language like Python, where the answer could be computed by executing the code. While such CoTs would be highly fragile to perturbation (due to syntax errors), this approach would not generalize to general language modeling tasks like Wikipedia text prediction where the "answer" cannot be computed deterministically.

2. **Question-CoT Pairs:** We could maintain the standard approach of keeping both question and CoT in context when predicting answers, measuring how perturbations to the CoT affect predictions. However, this makes it impossible to isolate whether the observed fragility stems from the CoT itself or from the interaction between question and CoT.

3. **Minimal Prompted CoTs:** We could prompt the model to produce minimal CoTs and measure their fragility to perturbation. This baseline is effectively represented at training step 0 in Figure 7, where we see minimal difference in log probability between perturbed and unperturbed CoTs from the pre-trained model.

Each of these potential baselines has significant limitations that prevent direct comparison with our approach. The formal language approach sacrifices generality, the question-CoT approach introduces confounding variables, and the minimal prompted approach is already captured as the starting point of our training process.

This analysis suggests that establishing meaningful baselines for faithful reasoning remains an open challenge in language model interpretability. Our approach provides one concrete formalization and optimization target, but we acknowledge there may be other valuable perspectives on what constitutes faithful reasoning that could lead to different baseline approaches in future work.

## F  CASE STUDY OF WIKIPEDIA PREDICTION

To better understand Llama-3.1-8B-Instruct's behavior after Markovian training to predict Wikipedia text, let's examine a its behavior on the (randomly selected near the end of training) batch number 12500. The model was given the following Wikipedia article excerpt:

> Boones Mill is a town in Franklin County, Virginia, United States. The population was 239 in 2018, down from 285 at the 2000 census. It is part of the Roanoke Metropolitan Statistical Area.
>
> History
> Boones Mill was incorporated in 1927. It was previously known as "Boone Mill" and "Boon Mill". The town is named after Jacob Boon who operated a mill in the town.
> The Boones Mill Norfolk & Western Railway Depot is listed on the National Register of Historic Places. https://www.townofboonesmill.org/history
> Geography
> Boones Mill is located in northern Franklin County at (37.115462, -79.953966), along U.S. Route 220 at the southern base of the Blue Ridge Mountains. US 220 leads north to Roanoke and south to Rocky Mount, the Franklin County seat.
> According to the United States Census Bureau, the town has

Given this context ending with "According to the United States Census Bureau, the town has", we can compare the CoT generated by our trained versus an untrained model:

### F.1  CoT after Training:

> "The town of Boones Mill is located in Franklin County, Virginia, United States. US 220 leads north to Roanoke and south to Rocky Mount, the Franklin County seat. According to the United States Census Bureau, the town has"

### F.2  CoT before Training:

> "The town of Boones Mill is a small, rural community with a rich history and natural beauty. The town is surrounded by the Blue Ridge Mountains, offering scenic views and outdoor recreational opportunities. The town's economy is primarily based on agriculture and small"

### F.3  Actual Continuation:

> "a total area of , all of it land. The town is in the valley of Maggodee Creek, a southeast-flowing tributary of the Blackwater River, part of the Roanoke River watershed. Murray Knob, elevation , rises to the north on the crest of the Blue Ridge, and the eastern end of Cahas Mountain, at , is 2 miles to the west."

The trained CoT shows notably different characteristics from the untrained one. The trained CoT essentially copied the first and last two sentences from the context, making sure to line up the number of allotted tokens with the end of the last sentence. The untrained model seems to give fairly generic properties that the actual Boones Mill Wikipedia article does not mention, such as Boones Mill having an economy primarily based on agriculture. Also, the untrained CoT is not taking the token limit into account and is setting the evaluator model to be surprised when it glues the CoT to the answer and has to predict "agriculture and small a total area of , all of it land".

This example achieved a normalized reward of 0.3438 (in log probability), suggesting that the trained CoT strategy was indeed helpful for predicting the technical geographic description that followed.

