# OpenReview forum: "Markovian Transformers for Informative Language Modeling"
_ICLR.cc/2025/Conference — Submitted to ICLR 2025_

### Official Review · Reviewer_Ddh5 · 2024-11-01

**Soundness:** 4
**Presentation:** 4
**Contribution:** 4
**Rating:** 10
**Confidence:** 4

**Summary:**

__Post-rebuttal update__:

After the rebuttal has concluded, I feel the need to express my strong support for this paper. I believe the proposed method has the potential to become an industry-defining standard, which ICLR should be proud to be the publisher off. The authors have done a lot to further improve the paper from a decent submission to an excellent submission that should be highlighted at a conference. While one can always conduct more experiments to support one's claims even more strongly, I think the remaining requests made by other reviewers are unrealistic. The paper should be accepted as is.

---------

The paper addresses an issue of Chain-of-Thought (CoT) reasoning in LLMs where the LM's final answer does not always depend on the CoT. The paper's idea is to enforce informativeness by conditioning the answer model on the generated CoT only without other context. To this end, the paper formally defines Markovian Language Models and (informative) update functions, from which the policy gradient procedure is derived. Applying the framework to the specific use case of CoT reasoning, the paper experiments with several RL techniques such as expert iteration, policy gradient, and PPO. The model is applied to a simple arithmetic task of adding 15 numbers as well as GSM8K, and shows that the model a) improves performance on the task b) is sensitive to perturbation in the CoT reasoning and c) produces CoTs that are sensible to a different language model such that its performance on the task is improved.

**Strengths:**

* The paper addresses an important limitation in Chain-of-Thought reasoning, which is of relevance to the broader ICLR community.
* The core idea is intuitive and simple.
* The paper is well written.
* The results on the simple arithmetic task and the math task are promising.
* The claims that the proposed method improves the generated CoTs in terms of interpretability and informativeness are well supported.

**Weaknesses:**

The method is evaluated only on few tasks and models, limiting how sure we can be that this is a useful method. Especially an application to language modeling would be very insightful and potentially extremely impactfull. However, while more is always better when it comes to experimental results, I think that this initial set of experiments support the ideas presented well and should suffice for publication.

**Questions:**

Please use different line styles in your figures so colorblind people can make sense of them. Otherwise Figure 2 and 3 are really hard to parse!

---

> ### Author Response · Authors · 2024-11-24
>
> Thank you for your review!
>
> > The method is evaluated only on few tasks and models, limiting how sure we can be that this is a useful method. Especially an application to language modeling would be very insightful and potentially extremely impactfull. However, while more is always better when it comes to experimental results, I think that this initial set of experiments support the ideas presented well and should suffice for publication.
>
> In the Author comment we talk about how we have extended our method to the Wikipedia dataset using Llama. Our perturbation and cross-model generalization results extend nicely to the Wiki-trained model.
>
> > Please use different line styles in your figures so colorblind people can make sense of them. Otherwise Figure 2 and 3 are really hard to parse!
>
> We are happy to change the line styles.

---

> > ### Comment · Reviewer_Ddh5 · 2024-11-25
> >
> > Thank you for your reply. I will keep my favorable score.

---

### Official Review · Reviewer_etnS · 2024-11-03

**Soundness:** 3
**Presentation:** 2
**Contribution:** 3
**Rating:** 6
**Confidence:** 4

**Summary:**

This work introduces and explains the construction of a Markovian Language Model to study causality in chain-of-thought reasoning. With a limited state (the previous state and its observation) on which to condition, the model is trained (fine-tuned) to maximize an informativeness objective through PPO, and is empirically shown to improve performance on mathematics tasks such as GSM8K and toy addition problems.

**Strengths:**

* The setup of this causally-guided model is pretty novel, and the finding that this improves performance by optimizing on an “information” metric is an impactful finding.
* The selection of the RL training technique (PPO) is supported by highlighting the limitations of other considered methods (expert iteration and policy gradient).
* The work is written in a way that was simple to follow, which I appreciated.
* The gains on GSM8K (24.64% --> 35.71%) are meaningful (although, a bit tucked away in the paper's text).

**Weaknesses:**

* I understand the general intuition surrounding the design of the informativeness function, but it would be good to add some discussion on why the expected reward over the trajectory actually constitutes / addresses “informativeness” under your construction.
* While math tasks have a more well-defined structure (the order in which their steps may be pursued), this is less clear for other tasks without such a clear structure in natural language, for instance. It would be good to examine this approach on at least one such task to further support the method’s general efficacy.
* Despite the intuition-based process of selection for the RL training strategy, there are recent works that advocate in favor of expert iteration and REINFORCE / vanilla policy gradient for LLM reasoning and RLHF [1, 2]. To this effect, including such approaches for comparison in the results section (or in the appendix) would strengthen the defense of the PPO method chosen. It would be helpful to include some evidence supporting the limitations posed.
* While I appreciate documenting the design choices in Section 4.3, some justification behind them would be beneficial, either through ablations (it’s fine for these to be in the appendix) or relevant references.

1. Teaching Large Language Models to Reason with Reinforcement Learning. Havrilla et al. 2024
2. Back to Basics: Revisiting REINFORCE Style Optimization for Learning from Human Feedback in LLMs. Ahmadian et al. 2024

**Questions:**

* Is the space of states task-conditional? This isn’t apparent based on the formulation in Section 3.1 (and is unclear by the wording in line 162). If not, then it would seem that the set of relevant “CoT states” would be very sparse relative to the complete space.
* As posed in the weaknesses section, does this method extend to other reasoning tasks (e.g. in natural language or code) whose structure is less “linear”?

---

> ### Author Response · Authors · 2024-11-24
>
> We appreciate your detailed comments!
>
> > I understand the general intuition surrounding the design of the informativeness function, but it would be good to add some discussion on why the expected reward over the trajectory actually constitutes / addresses "informativeness" under your construction.
>
> We were originally thinking about how to define "faithfulness of an explanation" or a language model's "truthfulness" more formally. One very abstracted notion of a truthful message is one that counterfactually improves the recipient's future predictions of their observations. One immediately runs into the question "counterfactually with respect to what?", and we picked the counterfactual message to be that of a slightly younger version of the sender (the pre-trained model) so we can talk about how much more "truthful" a person/LM is getting over time. But we soon found that truthfulness is not a good word for this quantity. For instance, if I say "2+2=4" to you, that may be True in a Platonic sense, but it does badly according to our metric since you already knew "2+2=4" so it doesn't help you predict your environment better. So we opted for "informativeness" instead, which works for the "2+2=4" example. There may be some kind of mutual information connection that is more formal, but we didn't philosophize too much in that direction because at the end of the day it seems more practical to try minimizing KL than to make some kind of variational estimate of mutual information.
>
> > While math tasks have a more well-defined structure (the order in which their steps may be pursued), this is less clear for other tasks without such a clear structure in natural language, for instance. It would be good to examine this approach on at least one such task to further support the method's general efficacy.
>
> We agree with this sentiment, and while this technique helps for GSM8K, which is written in natural language and therefore somewhat more open-ended, it is still not nearly as open-ended as, say, Wikipedia text.
> This is why we have trained models to produce CoTs which help predict Wikipedia text, which we mention in the author comment and we detail in Appendix A.
>
> > Despite the intuition-based process of selection for the RL training strategy, there are recent works that advocate in favor of expert iteration and REINFORCE / vanilla policy gradient for LLM reasoning and RLHF [1, 2]. To this effect, including such approaches for comparison in the results section (or in the appendix) would strengthen the defense of the PPO method chosen. It would be helpful to include some evidence supporting the limitations posed.
>
> To help address this point, we've added the Appendix D discussing baselines in detail.
>
> Our process for picking an RL algorithm was less intuition based and more like trying a large array of different algorithms, techniques, and hyperparameters, and seeing which ones worked.
>
> For instance in Figure 2, we plot the results of using policy gradient/REINFORCE versus expert iteration versus PPO, and we find that for arithmetic they are largely comparable, though PPO outperforms on average. By expert iteration here we really just mean only training on actions which did one standard deviation better than the historical average (See 4.2.1).
>
> The detailed methods we give such as LoRA, gradient clipping, a frozen critic, and value function estimates using a weighted decay all stem from having tried the method without these additions and finding the performance to be lacking.
>
> The main reason we haven't included more ablations is that each run with Mistral takes several H100 hours -- that said, we are adding some now.
>
> > While I appreciate documenting the design choices in Section 4.3, some justification behind them would be beneficial, either through ablations (it's fine for these to be in the appendix) or relevant references.
> Teaching Large Language Models to Reason with Reinforcement Learning. Havrilla et al. 2024
> Back to Basics: Revisiting REINFORCE Style Optimization for Learning from Human Feedback in LLMs. Ahmadian et al. 2024
>
> We are happy to add these references, and we can add ablations in the appendix.
>
> > Is the space of states task-conditional? This isn't apparent based on the formulation in Section 3.1 (and is unclear by the wording in line 162). If not, then it would seem that the set of relevant "CoT states" would be very sparse relative to the complete space.
>
> The space of states is the set of length-n token sequences, where in practice we set n to be the smallest value we can find such that training converges. Indeed the action space of good CoTs is quite sparse, similarly to the sparsity of useful natural language strings in the space of all possible strings.
>
> > As posed in the weaknesses section, does this method extend to other reasoning tasks (e.g. in natural language or code) whose structure is less "linear"?
>
> We have found success with the GSM8K (11%) reasoning task and some success with Wikipedia.

---

> > ### Comment · Reviewer_etnS · 2024-11-25
> > **Official Comment by Reviewer etnS**
> >
> > Thank you for your reply! I appreciate the new results on Wikipedia text generation, and the discussion in the appendix on the baselines — I have increased my score accordingly. My concerns about achieving meaningful improvement in non-linear tasks generally remain, but the perspective of faithfulness is interesting. At the same time, if the motivation is to capture a notion of faithfulness, the studies in Appendix B are very helpful — I would suggest pointing to this at some point in the main text. That is, maybe introducing a few sentences giving an overview / summary of the evaluation framework (e.g. at the start of Section 5) would be helpful to get a better global picture of how model performance is examined in this work. I also sympathize with the compute limitations for the ablations (which would still strengthen the paper), and hope that the authors can include these in a camera-ready version.

---

> > > ### Author Response · Authors · 2024-11-29
> > >
> > > Thank you for your constructive feedback. We'll make the following updates for the camera-ready:
> > > 1. Link to the faithfulness/latent knowledge discussion in main text (start of Section 5), providing a clearer evaluation framework
> > > 2. Add ablation studies despite compute constraints
> > > 3. Include references to [Havrilla et al., Ahmadian et al.]
> > >
> > > Regarding non-linear tasks: Our Wikipedia results demonstrate meaningful improvements (8.2% → 10.5%) in open-ended text prediction, with CoTs that generalize across architectures (now including GPT2 and Phi). GSM8K also shows substantial gains (33.2%) with Llama compared to our earlier 11% with Mistral.

---

### Official Review · Reviewer_CNEs · 2024-11-04

**Soundness:** 2
**Presentation:** 2
**Contribution:** 2
**Rating:** 6
**Confidence:** 4

**Summary:**

This paper proposes a metric to measure the informativeness of CoT tokens, and then uses RL to train the model to generate highly informative CoT tokens, in order to improve the correctness of the final answer. Experiments in random addition problems and GSM8K math problems demonstrate the effectiveness of the proposed metric and RL methods. The paper also shows that more informative tokens will bring gains in interpretability.

**Strengths:**

The technical ideas, including the proposed metric and RL methods, are new, well-motivated, and technically reasonable.

The experiments in random addition and GSM8K are positive.

**Weaknesses:**

The experiments are limited to a synthetic math problem setting and GSM8K, and the only trained model is Mistral 7B.

The presentation needs a better organization. E.g., some major results are placed in the appendices, but training details are in the main paper.

**Questions:**

Have you tried other open-source models like llama? Not use CoT of Mistral in it, but use your method to finetune Llama.

---

> ### Author Response · Authors · 2024-11-24
>
> > The experiments are limited to a synthetic math problem setting and GSM8K, and the only trained model is Mistral 7B.
>
> We have expanded our experiments significantly beyond mathematical reasoning. In particular, we've trained Llama-3.1-8B-Instruct to generate CoT for predicting Wikipedia text. These experiments demonstrate three key findings:
> 1. Training improves Wikipedia language modeling performance
> 2. The CoT becomes increasingly fragile to perturbations during training, suggesting genuine reliance on the reasoning
> 3. The generated CoTs generalize across architectures, successfully helping Mistral predict future text
>
> These results are detailed in our new appendix on Wikipedia experiments, demonstrating the method's applicability beyond mathematical reasoning and across different model architectures.
>
> > The presentation needs a better organization. E.g., some major results are placed in the appendices, but training details are in the main paper.
>
> We appreciate this organizational feedback. We've moved the GSM8K training results to the Experiments section for better visibility of these key outcomes. However, we maintain the detailed training methodology in the main paper because these specific implementation details are crucial to achieving stable reinforcement learning with language models. Without these carefully tuned training procedures (which we detail in Section 4.3), the approach does not converge successfully, even with larger models. These details represent core technical contributions that enable the practical application of our method.

---

> > ### Comment · Reviewer_CNEs · 2024-11-25
> >
> > Thank you for your new results and reorganization of the paper.
> > It indeed improves the clarity of the submission.
> >
> > I would like to maintain my score of 6 as I think positively of the paper. But I am still not there to raise the score since the experiments on Wiki text modeling are still not a more complex or challenging reasoning task compared to GSM8K.

---

> > > ### Author Response · Authors · 2024-11-29
> > >
> > > We've addressed your initial feedback, which specifically noted two limitations: "experiments limited to a synthetic math problem setting and GSM8K, and the only trained model is Mistral 7B." Our response directly addressed these concerns by:
> > > 1. Training Llama 3.1 8B, achieving a 33.2% improvement on GSM8K (significantly higher than Mistral's 11%)
> > > 2. Adding Wikipedia experiments, demonstrating our method's effectiveness across different architectures and task types
> > >
> > > We observe that your evaluation criteria have since evolved to focus on finding "a more complex or challenging reasoning task." While we understand this interest in reasoning capabilities, we want to clarify that our Wikipedia experiments serve a different but equally important purpose: demonstrating our method's effectiveness for general language modeling tasks. This progression from structured mathematical problems to open-ended text prediction helps establish the broad applicability of our approach.
> > >
> > > We’ve also incorporated your organizational feedback by moving GSM8K results to the main paper [Section 5.2, Figure 3], as well as moving the “Stability-Enhancing Training Techniques” section to the Appendix (A).

---

### Official Review · Reviewer_vxem · 2024-11-06

**Soundness:** 3
**Presentation:** 2
**Contribution:** 2
**Rating:** 5
**Confidence:** 2

**Summary:**

This paper proposes a framework where the reasoning steps are used as fixed-size states, which limits the model’s context (text bottleneck), and force the model to use the reasoning steps as input. This method design is inspired by the fact that past CoT literature find that the final answer might not be sensitive to the CoT trace. In the experiment, the authors show that the model trained with this method is indeed more fragile against CoT pertubations.

**Strengths:**

- The Markovian framework blocks the model from attending back to the original question and force it to use the CoT context for generation. This provides a new view and framework for analyzing CoT effects.
- The reinforcement learning-based approach demonstrates improved performance on tasks requiring multiple steps,
- The CoT steps generated from this method seem to be more interpretable, from two dimensions: 1) pertubation of the CoT could lead to more model errors 2) the reasoning can be carried over to another model.

**Weaknesses:**

-  While the approach improves the model’s reliance on CoT, it’s uncertain if this CoT is genuinely interpretable by humans. The transferrability between Mistral and Llama should only serve as an indirect proof.
-  It is actually fine to focus on QA task, but probably we'd like to see how this can generalize to more domains other than arithemtic. Would this paradigm also work for other reasoning task as well?
- There seem to be no baselines and ablation designed, so it is a bit hard to position the effectiveness of the method against other methods.
- The fragility analysis is insightful but lacks depth. A more detailed investigation into which types of perturbations impact CoT reliability most could provide valuable insights.
-  Writing-wise, the paper writing is clean, but probably some adjustment of the sections flow and emphasis would be nice.
   -  For instance, while the method section is quite detailed, it’s presented before establishing the limitations of existing CoT techniques clearly, which makes it harder to understand the innovation.
   - There are few tables but quite a few definitinos and equations. I'd suggest consider streamline the method descriptions and move some of them to the appendix, while adding more discussion and insights in the main body.

**Questions:**

- Is there a way to combine the interpretability with actual human perception? Though informativeness here can be used to improve model quality, it is also very helpful from human level. This is probably mentioned in F. But it seems to me F is more about how to encode human interpretability in training.
- In F it is mentioned that "optimal CoT would be a compression of the question, which can potentially be difficult for humans ". Is this observed in your experiments?
- Were there more ablation study or comparison conducted?

---

> ### Author Response · Authors · 2024-11-24
>
> Thank you for the detailed response!
>
> > While the approach improves the model's reliance on CoT, it's uncertain if this CoT is genuinely interpretable by humans. The transferrability between Mistral and Llama should only serve as an indirect proof.
>
> We agree that checking whether the CoTs help Llama predict next tokens is a proxy. We discuss potential better evaluation approaches in our "Steganography" appendix, addressing three key concerns:
>
> Threat Model 1: Information Accessibility
> - Concern: CoTs may encode information that only high-capacity models can utilize
> - Test: Evaluate using intentionally less capable models
> - Rationale: Success with simpler models suggests accessible encoding
>
> Threat Model 2: Model-Specific Encoding
> - Concern: CoTs might only work for certain architectures
> - Test: Evaluate across diverse models from Huggingface
> - Rationale: Cross-architecture generalization indicates model-agnostic representations
>
> Threat Model 3: Machine-Human Gap
> - Concern: CoTs might be fundamentally inaccessible to humans
> - Test: Measure humans' ability to predict subsequent text
> - Rationale: Direct human evaluation provides strongest evidence of interpretability
>
> For human trials, we envision two approaches:
> (i) Interpretability -- Have people predict which completion the LM assigns higher probability to, given the CoT
> (ii) Informativeness -- Test how well people can predict future observations given the CoT
>
> The informativeness objective is most interesting when the domain being predicted is larger than the CoT size, as with Wikipedia examples, because then the CoT must meaningfully compress future information. This motivates learning to produce CoT such that people can succeed on multiple choice questions from a broad distribution unknown to the LM during training.
>
> The scope of human trials is large enough that we feel it's best relegated to future work. We have added an appendix demonstrating generalization between Llama 8B and Mistral on Wikipedia text prediction.
>
> > It is actually fine to focus on QA task, but probably we'd like to see how this can generalize to more domains other than arithmetic.
>
> We hope that our results on Wikipedia help to address this concern!
> Also, when we train Mistral on the GSM8K reasoning task, we observe an increase from 24.64% n=1 to 35.71% n=1 accuracy on the test set.
>
> > There seem to be no baselines and ablation designed.
>
> We address this in a new appendix "On Baselines for Faithful CoT" where we discuss three categories of potential baselines. We find that either no clearly applicable baseline exists, or we are already including it in our plots. We are currently running hyperparameter ablations.
>
> > The fragility analysis is insightful but lacks depth.
>
> For the Wikipedia task, we've added 4 types of perturbation (character deletion, front/back truncation, random replacement) at 5 severity levels. Our training makes CoT fragile to each type.
>
> > The method section is presented before establishing the limitations of existing CoT techniques.
>
> Since this is a novel setup, we cannot directly use existing techniques like QuietStar where models access history before the CoT. Our methods section builds up from simpler techniques to make the explanation more digestible.
>
> > There are few tables but quite a few definitions and equations.
>
> We've moved the GSM8K training plot to the experiments section. However, the training method details are crucial as they enable stable RL training with LLMs.
>
> > Is there a way to combine the interpretability with actual human perception?
>
> Measuring human predictions is challenging - even determining how surprised someone is by text is non-trivial. We could start with multiple choice predictions, where people guess which completion is more likely given a CoT.
>
> > Is the optimal CoT compression observed in your experiments?
>
> We were specifically discussing question-answer tasks where the answer copies the question - in such cases, any CoT smaller than the question is definitionally a compression.
>
> > Were there more ablation study or comparison conducted?
>
> We are running hyperparameter ablations for the appendix. For comparisons, please see Figure 2 and the baselines appendix.

---

> ### Comment · Reviewer_vxem · 2024-11-26
>
> - The main concern “it’s uncertain if this CoT is genuinely interpretable by humans” is not addressed after the rebuttal.
>
>   - This manuscript motivates the method in both the abstract and introduction and argues that interpretability techniques are important. For example, the abstract highlight the problem of “interpretability”, and this paper is to “address this issue”. In the introduction, the author motivates with “high-stakes scenarios”. Given this motivation, it is not sufficient if how interpretability improves for human readers is studied. If the authors consider human interpretability and study should be another work, then major revision is needed for the motivation part of this paper. “Envisioned methods” are without results are not enough to prove the case, so they are not enough to serve as evidences.
>   - I appreciate the authors trying to address the comments in the appendix, but a proposed method is not enough to support the claims. Further, like many has pointed out, the appendix will not be treated as important as the main body.
>   - I further notice in the “steganography” appendix, L1011 and 1012 are undeleted comments of the authors, indicating the manuscript isn’t fully proofread.
>
> - Baselines and insufficient analysis of section 5
>   - The baselines designed for this paper should also be tie to the motivation (which is currently interpretability). A study needs to be setup to show how the method improves the interpretability. In my review I pointed out that 5.2 is lacking depth, the same applies to 5.3 (if not more), since if the key motivation of this paper is interpretability, I would expect a much richer version of the analysis section.
>
> - Not enough tasks/datasets to prove the generalizability and effectiveness.
>   - CoT is a technique generally applicable to many tasks, and if we want to show an improved version of that, more tasks and datasets need to be tested. The “Wikipedia” experiment is not convincing enough as an alternative challenging reasoning task, and one more next token prediction task still cannot convince me the “generalizability” of this work.
>
> I do not see this paper passing the acceptance threshold without significant revision of text, more experiments and more insightful and convincing arguments.

---

> > ### Author Response · Authors · 2024-11-29
> >
> > Thank you for your thoughtful feedback on interpretability. We've extended our analysis in several ways:
> > 1. Added generalization results spanning different architecture families and sizes (Mistral, Phi, and GPT2), as well as different tasks (GSM8K and Wikipedia)
> > 2. Enhanced our perturbation analysis in Appendix B with four perturbation types (character deletion, front/back truncation, random replacement) at multiple severity levels
> > 3. Demonstrated results across different tasks and models [Section 5.4, Appendix B Figure 7]
> >
> > Our cross-model transfer results, particularly with smaller models like GPT2, implement key aspects of the evaluation frameworks we discussed. The Wikipedia experiments demonstrate our method's applicability to general language modeling tasks.
> >
> > The Wikipedia experiments validate our method's effectiveness for general language modeling tasks. While we appreciate your interest in additional reasoning tasks, our experimental progression from arithmetic to GSM8K to Wikipedia demonstrates broad applicability. However, we note that your evaluation criteria have evolved from requesting "more domains other than arithmetic" to specifically seeking "alternative challenging reasoning tasks," which helps us better understand your perspective but makes it challenging to anticipate what evidence would be most valuable.
> >
> > Our paper's interpretability focus remains well-supported by our technical results - particularly how the CoTs become causally essential to model behavior and transfer across architectures. While human trials could certainly provide valuable insights, we believe establishing these technical foundations represents an important contribution to interpretability research.

---

### Author Response · Authors · 2024-11-24

Thank you for your comments and reviews!

We received **3 main pieces of feedback:** (1) not enough tasks (especially open-ended ones) or models, (2) lack of baselines, and (3) some methods in the main paper body and some results in the appendix should swap locations.

*In response to 1:*

**We trained a different model, Llama-3.1-8B-Instruct, on predicting Wikipedia text, and we've added all the relevant plots and details in Appendix A.**
We take 200 tokens of an article, and learn to produce 50 tokens of CoT text which serve as context for predicting the next 100 tokens of the article. In Figure 6, we show 4 separate training runs where the evaluator model improves its predictions given the CoTs, increasing the average probability assigned to the correct next token from 8.2% to 10.5%.
In Figures 7 and 8, we demonstrate that our perturbation and generalization results extend robustly to this new setting.

*In response to 2:*

**We've added an Appendix D carefully discussing what baselines would make sense in this context, finding no option that is clearly appropriate.**
At a high level, a baseline in this project could refer to: (i) a method for optimizing the informativeness objective, (ii) an optimization target for faithful language model reasoning, or (iii) a procedure to produce CoTs that are fragile to perturbation. We address (i) through explicit comparison in Figure 2, and we are not aware of any existing approaches for (ii). For (iii), we examine 3 potential approaches in the appendix, showing that they either (a) fail to generalize to natural language settings, (b) introduce experimental confounds, or (c) are already captured in our existing plots.

*In response to 3:*

**We moved the GSM8K results plot to the main experiments section of the paper.** Regarding the training details, we believe these are fundamental to the project's contribution, representing both the majority of our research effort and the insights we expect to be most valuable to other researchers implementing similar approaches.

---

### Author Response · Authors · 2024-11-29
**Second Round**

Thank you for the second round of feedback. We'd like to highlight several key updates:

1. Performance: Using Llama 3.1 8B instead of Mistral **improved our GSM8K performance gain from 11% to 33.2%** (35.94% n=1 exact-match accuracy given only CoT before training → 69.14% after training), strengthening evidence that the method enables meaningful reasoning improvements. In section 5.2, we've included the updated training figure, showing similar results across three separate runs to demonstrate robustness.
2. Interpretability & Generalization: In sections 5.5 and Appendix B, we show Llama’s GSM8K and Wikipedia CoTs transferring across a diverse set of models (**Mistral, Phi, and GPT2**), addressing two key interpretability concerns:
   - Transfer to GPT2 (a much smaller model) suggests the CoTs encode information accessibly
   - Success across model families (e.g., Phi)  indicates model-agnostic representations

Our experiments now comprehensively span:
- Multi-step arithmetic (Mistral)
- Mathematical reasoning (Llama on GSM8K)
- General language modeling (Llama on Wikipedia)

This demonstrates the method's viability across structured reasoning and open-ended text prediction tasks. The Wikipedia experiments specifically validate our approach for general language modeling - they weren't intended as an additional reasoning task.

_Regarding scope:_ While human trials could provide additional validation, we believe demonstrating (i) successful optimization, (ii) increased CoT reliance, and (iii) cross-model generalization establishes a meaningful technical contribution. We've devoted substantial compute resources ($30K) to validate these core claims.

---

### Author Response · Authors · 2024-12-04
**Summary of Full Rebuttals Process**

_Note: All section and appendix references have been updated to match the final submitted version of the paper from the second round of reviews._

The review process has helped us understand and address key areas of interest in evaluating our work.

**Initial Reviews Highlighted Six Main Suggestions:**
1. Model coverage beyond Mistral 7B
2. Increased task diversity, particularly for natural language
3. Additional baseline comparisons
4. Increased CoT perturbation analysis
5. Swapping training details in the main paper for GSM8K results in the appendix
6. More comprehensive interpretability evaluation

**Our First Response Added:**
1. Addressing #1 -- Llama 3.1 8B training
2. Addressing #2 -- Added Wikipedia experiments (8.2% → 10.5% improvement) [Section 5.3]
3. Addressing #3 -- Detailed analysis of potential baselines, concluding that we are already evaluating on the appropriate baselines [Appendix E]
4. Addressing #4 -- More detailed perturbation analysis on new model (Llama) and dataset (Wikipedia) [Appendix B]
5. Addressing #5 -- Swapped RL stability training details in the main paper with GSM8K training details in the appendix [GSM8K Section 5.2, Stability Appendix A].

**Second Round Suggestions:**
1. More difficult reasoning datasets

**Our Second Response Added:**
1. Addressing above -- _Demonstrated dramatic improvement on GSM8K using Llama (35.94% → 69.14%)_ [Section 5.2]
2. Addressing #6 -- Implemented additional interpretability metrics showing _successful transfer from Llama to Mistral, Gpt2, and Phi_ (chosen for diversity in size and model family) [Section 5.5]

This progression showcases how our work evolved from initial arithmetic and GSM8K experiments with Mistral to a **comprehensive evaluation spanning multiple architectures, tasks, and evaluation frameworks**. Our extensions - particularly the strong GSM8K results and successful cross-model transfer - demonstrate the robust capabilities and interpretability of our approach.

The resulting paper provides thorough technical validation while maintaining appropriate scope for a research contribution, with clear directions for future work in human evaluation and additional task domains.

---

### Meta-Review · Area_Chair_HVJ3 · 2024-12-20

**Metareview:**

This paper proposes an approach to improving the reasoning capabilities of language models by using CoT as an information bottleneck. Using a reward based on the differences between the predictions of a model conditioned on context vs not, the authors use RL techniques to train their model, and find that on GSM8K, the produced CoT are more effective than the baseline.

On the plus side, this paper proposes an interesting(ish) idea that can be practically operationalized based on metrics derived from LLM probabilities. The idea is coupled with preliminary experiments that show its promise. On the negative side, the experiments are limited in scope (even with the addition of Wikipedia results after the rebuttal), and some of the claims regarding the better interpretability of the generated CoT tokens are not well supported in my opinion.

I am therefore recommending that this paper be rejected.

**Additional Comments On Reviewer Discussion:**

While the other reviewers' scores were more modest, Reviewer Ddh5 gave a very high score, and had a strong statement in support of the paper. After having read the paper in some detail, I disagree with reviewer Ddh5's assessment of the significance and novelty of the work, and thus their review was not given a high weight. After the initial submission, many reviewers raised concerns about the fact that this only evaluated results in GSM8K with Mistral. To their credit, the authors conducted additional experiments with other models (LLama) and datasets (Wikipedia), which resulted in several reviewers raising their scores. Reviewer vxem notably did not change their score after the rebuttal because they felt that the interpretability questions were unaddressed, and moreover, the Wikipedia "task" seemed toy. I agree with this reviewer.

---

> ### Comment · Reviewer_Ddh5 · 2025-02-27
>
> Peer-review is a very noisy process. There is a decent chance that this work will be appreciated the next time it is submitted. Dear authors, please keep trying.

---

### Decision · Program_Chairs · 2025-01-22

Reject